# Mast Cells and Basophils in the Defense against Ectoparasites: Efficient Degradation of Parasite Anticoagulants by the Connective Tissue Mast Cell Chymases

**DOI:** 10.3390/ijms222312627

**Published:** 2021-11-23

**Authors:** Zhirong Fu, Srinivas Akula, Anna-Karin Olsson, Jukka Kervinen, Lars Hellman

**Affiliations:** 1The Biomedical Center, Department of Cell and Molecular Biology, Uppsala University, SE-751 24 Uppsala, Sweden; fuzhirong.zju@gmail.com (Z.F.); srinivas.akula@icm.uu.se (S.A.); 2Department of Medical Biochemistry and Microbiology, BMC, SE-751 23 Uppsala, Sweden; Anna-Karin.Olsson@imbim.uu.se; 3Tosoh Bioscience LLC., 3604 Horizon Drive, King of Prussia, PA 19406, USA; Jukka.Kervinen@tosoh.com

**Keywords:** basophils, mast cells, ectoparasites, ticks, leeches, mosquitos, anticoagulant

## Abstract

Ticks, lice, flees, mosquitos, leeches and vampire bats need to prevent the host’s blood coagulation during their feeding process. This is primarily achieved by injecting potent anticoagulant proteins. Basophils frequently accumulate at the site of tick feeding. However, this occurs only after the second encounter with the parasite involving an adaptive immune response and IgE. To study the potential role of basophils and mast cells in the defense against ticks and other ectoparasites, we produced anticoagulant proteins from three blood-feeding animals; tick, mosquito, and leech. We tested these anticoagulant proteins for their sensitivity to inactivation by a panel of hematopoietic serine proteases. The majority of the connective tissue mast cell proteases tested, originating from humans, dogs, rats, hamsters, and opossums, efficiently cleaved these anticoagulant proteins. Interestingly, the mucosal mast cell proteases that contain closely similar cleavage specificity, had little effect on these anticoagulant proteins. Ticks have been shown to produce serpins, serine protease inhibitors, upon a blood meal that efficiently inhibit the human mast cell chymase and cathepsin G, indicating that ticks have developed a strategy to inactivate these proteases. We show here that one of these tick serpins (IRS-2) shows broad activity against the majority of the mast cell chymotryptic enzymes and the neutrophil proteases from human to opossum. However, it had no effect on the mast cell tryptases or the basophil specific protease mMCP-8. The production of anticoagulants, proteases and anti-proteases by the parasite and the host presents a fascinating example of an arms race between the blood-feeding animals and the mammalian immune system with an apparent and potent role of the connective tissue mast cell chymases in the host defense.

## 1. Introduction

Ticks, lice, flees, mosquitos, leeches and vampire bats need blood as a food source and for their reproduction. One problem for these ectoparasites is that blood very easily coagulates when placed in contact with foreign material and several of these blood-feeding animals feed for a relatively long period, sometimes several days or weeks. These blood-feeding animals, therefore, produce potent anticoagulant proteins, potent complement inactivating proteins, anesthetics, immunosuppressives, profibrinolytics, and compounds that promote vasodilation [1,2,3,4,5]. Probably, the most well-characterized anticoagulant is hirudin, a potent inhibitor of thrombin consisting of 66 amino acids and produced by leeches. Hirudin acts by blocking the exosite 1 of thrombin [6,7]. The second family of anticoagulants, which also are thrombin inhibitors but with a unique inhibition mechanism, are the anophelins produced by different *Anopheles* mosquitos [8]. Anophelins also act primarily by blocking thrombin exosite 1, but with an opposite orientation of the peptide interaction between the anticoagulant and thrombin [8]. The third class of proteins are the anticoagulants produced by ticks. One such anticoagulant protein (TAP) has been isolated from the tick species *Ornithodoros moubata* [9]. TAP, which belongs to the Kunitz type family of protease inhibitors, inhibits the blood coagulation factor Xa [9].

The number of different tick species is vast and there is a very large diversity of anticoagulants among ticks. For example, among the *Ixodida* ticks the number of species is estimated to be ~900 species [10]. Among the *Haemaphysalis*, *Hyalomma* and *Rhipisephalus ticks*, several hirudin-type anticoagulants have been identified that also inhibit thrombin but with a different binding mode from hirudin by acting at exosite 2. These anticoagulants are called madanins and chimadanins [11]. These two thrombin inhibitors, with 60 and 70 amino acids, have two tyrosines in the exosite 2 interacting region that need to be sulfated for full activity [11]. Hirudin has one tyrosine in the exosite 1 binding region which also is important for its activity. The sulfated tyrosine in hirudin increases the inhibitory activity by one order of magnitude compared to the non-sulfated variant [11]. Another class of anticoagulants found in ticks is the variegin from the tropical bont tick *Ambyomma variegatum* [12]. This is one of the smallest anticoagulants so far identified with its 32 amino acid residues. Variegin is also a thrombin inhibitor acting on exosite 1 with high sequence identity in the region of hirudin interacting with exosite 1 of thrombin [12]. Separate from the low molecular weight anticoagulant proteins is Draculin of the common vampire bat, *Desmodus rotundus*. This bat species is found in Central and South America and exclusively feeds on mammalian blood [4,13]. Draculin acts on both factors IX and X in the blood coagulation cascade [13]. The anticoagulants described above for ticks, mosquitos and leeches are ~10 kDa or smaller whereas draculin is a complex glycosylated protein of ~85 kDa.

One potential way for the host to protect itself from these parasites is to inactivate its anticoagulants. To our knowledge, however, no such mechanism has conclusively been reported. Several of the major immune cells, including mast cells, neutrophils, cytotoxic T cells, NK cells and, to a lesser extent, basophils produce and store massive amounts of primarily serine-type proteases in their cytoplasmic granules. Thus, one potential mechanism for the neutralization of these anticoagulating proteins is inactivation by host proteases.

Basophils have been shown to be involved in the defense against ticks by migrating and accumulating in relatively large numbers in the skin under the tick injection area [14,15,16]. This phenomenon has been observed in several species including cattle, rabbits, guinea pigs and mice [15]. The accumulation of basophils does not occur at the first encounter with the parasite but only after the second infestation [16]. This accumulation of basophils at the site of the tick bite has been shown to be dependent on IgE, the high-affinity receptor of IgE (FcεRI) and on IL-3 from skin resident CD4+ memory T cells [17]. Selective ablation of basophils has also been shown to result in loss of acquired protection against ticks in both guinea pigs and mice [15]. Ticks inject protease inhibitors during a blood meal and the amount of these protease inhibitors increases dramatically in the tick salivary glands following the initiation of a blood meal [18]. One such protease inhibitor, the serpin IRS-2, has been cloned from the tick *Ixodes ricinus* and analyzed for its expression [18]. By day two after the tick has attached to the skin, a nine-fold increase was observed for IRS-2 and this upregulation increased to 36-fold by day six [18].

The strong upregulation of serpins upon initiation of a blood meal indicates that host proteases are a major factor in the protection against blood-feeding parasites [18]. Interestingly, these serpins have quite remarkable selectivity for two hematopoietic serine proteases, chymase and cathepsin G [18]. Both of these proteases are, however, expressed by human mast cells and not by basophils [19]. Previously, we have cloned a serine protease specifically expressed in mouse basophils, the mouse mast cell protease-8 (mMCP-8) [20,21,22]. In contrast to most other mammalian hematopoietic serine proteases, mMCP-8 appears highly restrictive in its cleavage specificity. The only target so far identified has been mouse α-tubulin, which most likely is not its biologically most important substrate [23]. Interestingly, the first clue to the physiological role of mMCP-8 has been from a study involving the injection of recombinant mMCP-8 into mouse skin [23]. In this study, mMCP-8 induced an inflammatory response in the skin with increased microvascular permeability and leukocyte infiltration in a protease activity-dependent manner. This finding indicated a role of mMCP-8 in activating other immune cells in the area of basophil activation and also a direct effect on the vasculature. However, no direct target that would explain this response to active mMCP-8 has yet been identified. mMCP-8 is a member of the large family of hematopoietic serine proteases where other members of this family have been shown to have an array of different activities in both immunity, blood pressure regulation and general tissue homeostasis [24,25,26]. Of interest for the role of these proteases in our defense against various parasites are the mast cell proteases, including the chymase and a mast cell specific carboxypeptidase, CPA3, that have been shown to be of major importance for our defense against snake, scorpion and gila monster venoms [27,28].

The question we address here is if mast cell and basophil serine proteases are directly involved in the defense against ectoparasites. Our experimental approach was to analyze the activity of multiple mast cell and basophil proteases on various anticoagulant proteins from ectoparasites and to study the role of the tick serpin in inhibiting these potential activities. We show that the connective tissue mast cell chymases, but not the closely related mucosal mast cell chymases, efficiently cleaved all three tested anticoagulants. These results suggest an important role for the mast cell-specific chymases in our defense against ectoparasite infections. We also show that mMCP-8 selectively cleaves MIP-3a and PDGF-B, indicating a feedback loop on the epithelial hyperplasia and potential fibrosis induced by the tick saliva components at the site of tick infestation.

## 2. Results

### 2.1. Production of a Recombinant Tick Anticoagulant Protein and Cleavage Test by a Panel of Hematopoietic Serine Proteases

As the first step in the search for a potential role of the different hematopoietic serine proteases in the defense against ectoparasites, we analyzed their activity against Tick Anticoagulant Protein (TAP), a potent anticoagulant protein expressed by the tick *Ornithodoros moubata* [9]. The coding region for TAP, a protein belonging to the Kunitz type of protease inhibitors, was ordered from Genscript as a designer gene and inserted in a vector pET21a containing an *Escherichia coli* thioredoxin gene as a fusion partner [9]. We produced this anticoagulant protein as a fusion protein with bacterial thioredoxin (Trx), because of its small size and the fusion with a stably folded bacterial protein often enhances correct folding. In the region between the Trx and the tick anticoagulant protein, we also inserted a His_6_ affinity tag, and an enterokinase site, to be able to purify the protein using Ni^2+^ chelating sepharose beads and to release the anticoagulant protein from the fusion partner after purification by enterokinase cleavage (Figure 1A,B). Following the production in a bacterial host (*E. coli* rosetta gami) the anticoagulant protein was cleaved with enterokinase and the part of the fusion protein containing Trx, His-tag and enterokinase site were removed from the solution with Immobilized Metal Ion Chromatography (IMAC) chelating Ni^2+^ beads. The recombinant tick anticoagulant protein was then subjected to cleavage by a panel of 17 recombinant mammalian mast cell, basophil and neutrophil proteases including mMCP-8 (Figure 1C). mMCP-8 had no activity on the tick anticoagulant protein (Figure 1C). However, all the connective tissue mast cell chymases from human (Hum. Chy.), dog (Dog Chy.), rat (rMCP-1), hamster (Ham. Chy.) and opossum (Opossum Chy.) and the human neutrophil cathepsin G (Hum. CG) cleaved the tick anticoagulant protein efficiently (Figure 1C). The striking result also was the complete absence of cleavage by the mucosal mast cell enzyme (rMCP-2), the rat vascular chymase and by the human neutrophil protease, proteinase 3. This shows that, almost exclusively, only the connective tissue mast cell enzymes have the specificity for the tick anticoagulant protein degradation. The only connective tissue proteases that did not cleave the tick anticoagulant protein were the two platypus enzymes (Platypus GzmB and Platypus DDN1) and the mast cell enzymes from rabbit and guinea pig and (Rabbit Chy. and Guinea pig Chy.) (Figure 1C). The guinea pig and rabbit enzymes have no chymotryptic activity but are strict Leu-ases, indicating that the cleavage of this tick anticoagulant protein is dependent on chymotryptic-type enzyme activity [29].

### 2.2. Production of the Anticoagulant Proteins from Mosquito and Leach and Analysis of Their Protease Sensitivity

To study the more general aspects of the role of the hematopoietic serine proteases in the defense against blood-feeding parasites, we decided to also test the activity of the above panel of hematopoietic serine proteases on the potent anticoagulants produced by the malaria vector *Anopheles gambiae*, a mosquito, and of the leech *Hirudo medicinalis*. These two proteins, anophelin and hirudin, respectively, were produced in *E. coli* similarly to the tick anticoagulant protein TAP, but without the Trx fusion partner. An N-terminal His_6_ tag was added for purification and an enterokinase site to be able to remove the His-tag and obtain the native protein.

Before analyzing the protease sensitivity of hirudin and anopheline, both recombinant anticoagulants were tested for their inhibitory capacity against thrombin and found to be highly potent in preventing the proteolytic activity of thrombin (Figure 1D,E). Interestingly, anophelin was fully active even in the presence of the His-tag and the enterokinase site, whereas hirudin was only active after removal of the His-tag and the enterokinase site (Figure 1D,E).

The recombinant hirudin and anophelin were then tested for their sensitivity to cleavage with a panel of mammalian hematopoietic serine proteases, similar to what was performed for the tick anticoagulant protein. The results show that both anticoagulants, similarly to the tick counterpart, were very sensitive to cleavage by the connective tissue mast cell chymases (Figure 2). In line with the data from the tick anticoagulant, no cleavage of hirudin and anophelin was observed by the mucosal mast cell proteases, neither by the mast cell tryptases nor by mMCP-8 (Figure 2). The visualization of the result for anophelin on SDS-PAGE was somewhat challenging as the anophelin is highly negatively charged, which explains why the staining is very weak (Figure 2B).

### 2.3. The Role of Tick Serpin IRS-2 in the Inactivation of Hematopoietic Serine Proteases

The tick serpin IRS-2 has previously been shown to be able to relatively specifically inactivate the human mast cell chymase and human cathepsin G [18]. To test the ability of this serpin to inactivate different hematopoietic serine proteases-and to analyze the possibility that some of these proteases could be involved in the inactivation of the serpin by cleavage-we produced the tick IRS-2 serpin as a recombinant protein in mammalian cells using the episomal vector pCEP-Pu2 and the human embryonic cell line HEK293-EBNA as host. After transfection and establishing a confluent culture, we purified the tick serpin from the conditioned media by affinity chromatography on Ni^2+^ chelating sepharose.

We first tested the activity of the serpin on human mast cell chymase using a titration of increasing amounts of chymase but keeping the serpin concentration constant throughout the experiment. The serpin was found to first form a covalent complex with the human chymase under conditions when the serpin is in molar excess (lane 16:17) (Figure 3A). When the chymase concentration was increased, this high molecular weight complex disappeared and instead, a partially digested complex appeared (lanes 24:17–60:17), which size was smaller than the serpin alone (lane 0:17) (Figure 3A). The serpin is a covalent irreversible inactivator, why each inhibitor molecule inhibits only one protease molecule. The serpin needs to be in molar excess to fully prevent the chymase activity. Conversely, when chymase was added in excess, the free chymase is active and cleaves the chymase-serpin complex (Figure 3A, lanes 24:17–60:17). This is shown in an activity assay where the chymase is active and cleaves a chromogenic substrate efficiently starting when the chymase is in molar excess (Figure 3B).

When tested on a larger panel of enzymes, we can see that most of the enzymes tested, including human chymase, dog chymase, rMCP-1, RVC, rMCP-2, hamster chymase, guinea pig Leu-ase, opossum chymase and all three human neutrophil proteases, cathepsin G, proteinase 3 and N-elastase, are inhibited by the serpin. However, the tick serpin had no effect on the rabbit Leu-ase, the human tryptase and mMCP-6, or against the basophil-specific protease mMCP-8 (Figure 3C,F). The inactivation of both hamster and opossum chymases by the tick serpin IRS-2 was verified by chromogenic substrate assays (Figure 3D,E), illustrating the good correlation between the gel-based activation assay and the chromogenic substrate assay.

### 2.4. Analysis of the Cleavage Specificity of mMCP-8

Since basophils accumulate at the site of tick infestation and mouse basophils express the basophil-specific serine protease mMCP-8, we hypothesized about its potential role in tick defense. Based on the results in Figure 1 and Figure 2, we conclude that mMCP8 is not capable of inactivating the anticoagulant proteins TAP, hirudin and anophelin. As mentioned above, the only protein that previously has been shown to be cleaved by mMCP-8 is mouse α-tubulin. However, tubulin is most likely not an important target for this enzyme as it is an intracellular protein. We, therefore, wanted to look deeper into its specificity. We started with phage display as this technique can give a detailed picture of the extended specificity of a protease [29,30,31,32,33,34,35,36,37,38,39,41,42,43,44,45,46,47]. We tried more than 10 times with the phage display technique and also with a large panel of both chromogenic and recombinant substrates to identify the cleavage specificity of mMCP-8, however, without success why we doubted if our protease was active. We, therefore, tested the cleavage by recombinant mMCP-8 on a panel of 52 different recombinant mouse cytokines and chemokines (Figure 4). Our recombinant mMCP-8 was found to cleave only two out of these 52 proteins, MIP-3a and PDGF-B (Figure 4A). This shows that the enzyme is active but extremely restrictive in its target specificity. The related PDGF-A was not cleaved by mMCP-8 (Figure 4B), which further indicates very high selectivity and target site selection by this enzyme. However, after analyzing the sequences of the two proteins cleaved by mMCP-8, MIP-3a and PDGF-B, we could not find a clear consensus site for cleavage, which was expected based on the apparent very high selectivity (Figure 4B,C). It is, therefore, possible that cleavage by the protease is not only dependent on the primary sequence but that the structural conformation around the cleavage site is also important, which could explain the high selectivity.

## 3. Discussion

Blood-feeding parasites are highly dependent on potent anticoagulants for success in their attempts to get a blood meal. The importance of anticoagulants for these parasites is strongly supported by the fact that the three species we analyzed here (tick, mosquito, leech) are coming from different parts of the animal kingdom and that the three anticoagulants analyzed here are very different in structure. These anticoagulants seem, therefore, not to have any common ancestor but most likely have appeared by convergent evolution. One way for the host to counteract these attempts to obtain blood can be to proteolytically degrade these anticoagulants. We here show that all classical connective tissue mast cell chymases, from human to opossum, effectively degrade these anticoagulants. An interesting finding is also that similar chymases from mucosal mast cells or vascular muscle cells, rMCP-2 and RVC, that have only slightly different cleavage specificity on linear substrates, have no or only minor activity on these anticoagulants (Figure 1 and Figure 2) [25,33,34]. Mucosal mast cells are positioned in the intestinal mucosa and in some animals also in the lung where there is no possibility for contact with these anticoagulant proteins. In contrast, the connective tissue mast cells are located in the skin, which is the site of contact with the blood-feeding parasite. This shows a remarkable selectivity in the evolution of the cleavage specificity of these mast cell proteases. An interesting finding was also the absence of cleavage of these anticoagulants by the platypus enzymes (Figure 1 and Figure 2). The platypus is primarily a water-dwelling animal that may have little contact with at least two of these anticoagulants, the tick and the *Anopheles* anticoagulants. Alternatively, there could also be another explanation to this phenomenon as the fauna of Australia and Tasmania is quite different from the rest of the world and that the ticks, mosquitos and leeches we have analyzed here may not be present in Australia and Tasmania, and that the platypus enzyme, therefore, has not had selective pressure for the inactivation of these anticoagulant proteins.

The fact that ticks produce protease inhibitors with potent activity against these mast cell enzymes is also a strong indication of the importance of these mast cell enzymes in the defense against these ectoparasites. It has been shown that the expression levels of the tick serpin IRS-2 increase in the tick salivary glands by 9-fold already by day 2 and by 36-fold by day 6 showing a strong correlation to active blood-feeding [18].

The blood-feeding process is a very complex process involving several important steps [4,5]. Inhibition of coagulation is most likely one of the first, inhibition of the degradation of the anticoagulants by injection of potent protease inhibitors may then come as the second step in this process. These protease inhibitors may, in addition to a role in blocking degradation of the anticoagulant, also have a direct role in the inhibition of the different coagulation proteases, such as thrombin and FX. IRS-2 has actually been shown to also act directly on thrombin but then at higher concentrations [18]. The presence of a good blood flow in the region of the skin of interest for the parasite seems also to be of major importance. A strategy for the parasite to increase access to blood is by injecting factors that widen and possibly cause leakage of blood vessels. Several examples of such compounds have been identified, including lipid mediators of the prostaglandin family [5]. Enzymes or other factors that enhance fibrinolysis seem also to be a common component of the saliva of the majority of these blood-feeding animals. Vampire bats produce a potent fibrinolytic factor, DSPA (*Desmodus rotundus* salivary plasminogen activator), closely related to the human plasminogen activator [4]. In ticks, both a fibrinolytic metalloprotease and a plasmin activator, Longistatin, have been identified [5,48] It is also well known that the saliva from these blood-feeding animals contains anti-inflammatory proteins. Among the most common are different complement inhibiting proteins, but also histamine-binding proteins [5]. Tick saliva components are often glycosylated and the proteins can thereby activate the mannose lectin pathway involving the MBL and MASP proteins. A family of low molecular weight such proteins have been identified in the *Ixodes scapularis* and the *Ornithodoros savignyi* ticks, the TSLPI (Tick salivary lectin pathway inhibitor), Salp14 and BSAP1 proteins [49]. Factors inhibiting the alternative pathway also seem commonly found among ticks [5].

Protease inhibitors are often found in the saliva of these blood-feeding animals, likely to prevent host degradation of the anticoagulants. We, therefore, wondered if the immune system has ways to counteract the effect of these protease inhibitors by protease cleavage. To address this question, we produced the tick serpin IRS-2 and tested it for its sensitivity to cleavage by the same panel of hematopoietic serine proteases as shown in Figure 1 and Figure 2. We found that this serpin had much broader activity than previously indicated. It efficiently inactivated almost all the connective tissue chymases from a panel of mammals including human, dog, rat, hamster and opossum, and also the mucosal mast cell proteases, the RVC and the neutrophil proteases. However, the mast cell tryptases and the basophil-specific protease mMCP-8 were not inactivated by this serpin. The lack of inactivation by IRS-2 indicated that these proteases are still active in the presence of this serpin and can play an active role in the defense against the ticks. Neither mMCP-8, nor the mouse and human tryptases seemed to cleave the serpin, why the role of both the tryptases and mMCP-8 may lay in other aspects of the defense. Human basophils express low levels of the tryptase, but there is no homolog to mMCP-8 in the human genome. Moreover, mouse basophils express a tryptase different from the mouse mast cell tryptases, mMCP-6 and 7 [26,50]. Mouse basophils instead express the tryptase mMCP-11 [50]. Thereby, both mouse and human basophils express a tryptase. Both human and mouse basophils also express the mast cell and basophil specific carboxypeptidase, the CPA3, showing clear similarities in protease repertoire between mouse and human basophils, despite the lack of a mMCP-8 homologue in human basophils.

Interestingly, injection of recombinant active mMCP-8 in the skin of mice has been found to induce an inflammatory response with increased microvascular permeability and leukocyte infiltration in a protease activity-dependent manner [23]. This indicates that mMCP-8, in addition to its cleavage of MIP-3a and PDGF-B, can have important activating functions both for leukocyte infiltration and vasodilation by cleavage of yet unknown targets. The cleavage of PDGF-B by mMCP-8 may have a dampening function on the epithelial hyperplasia that occurs at the tick attachment site. This hyperplasia may also include fibroblasts where the cleavage of PDGF-B could have a dampening function, similar to what appears to be the role of both the mast cell chymases and the tryptases on an excessive TH2 response by cleavage of a very selective number of TH2 promoting cytokines and chemokines [40,51]. This indicates that these enzymes have a very complex role in immunity and tissue homeostasis by simultaneously having both activating and dampening functions. The role of mMCP-8 in the cleavage of MIP-3a is also interesting. Does the N or C-terminal trimming of MIP-3a by mMCP-8 activate or inactivate this chemokine? The role of mMCP-8 in inducing and controlling inflammation will need further analysis before we know the full potential of this basophil specific enzyme. A similar question exists for another basophil specific mouse protease, the tryptase mMCP-11, which like mMCP-8 has been shown to induce leukocyte infiltration and vasodilation by cleaving yet unknown targets [50,52].

Interestingly, the connective tissue mast cell chymases had potent activity against the three anticoagulant proteins from ectoparasites, but not the proteases expressed by the basophils that often accumulate at the site of infection. The question is then how the mast cells, which do not increase in number at the feeding site, can get activated and contribute to the degradation of the anticoagulant proteins. It has been shown that IgE produced against various tick saliva proteins has an important role in the activation of the basophils that accumulate at the injection site [16]. This IgE does most likely also binds to mast cells in the area of infection. When the proteins from the saliva of ticks, mosquitos, and leeches slowly diffuse through the tissues it can most likely activate the mast cells in the area of infection which results in the release of the very abundant chymases, which in turn results in the inactivation of the anticoagulant proteins. The local mast cells can most likely also be activated by substance P and other positively charged compounds acting on the MRGPRX2 receptor or by complement components like C3a, C4a and C5a through the anaphylotoxin receptors.

A number of potential targets have been identified for the different mast cell proteases, including different connective tissue components, cell adhesion molecules, angiotensin I, thrombin, different venoms, and a highly restricted set of TH2 cytokines [24,28,33,34,40,51]. To this panel of targets, we can now add anticoagulant proteins of a panel of very diverse ectoparasites, such as ticks, mosquitos and leeches. This shows the very diverse role that these abundant mast cell enzymes have in immune defense and tissue homeostasis. The interesting question in the area of basophils and mast cells in the defense against ectoparasites is now the potential targets for mMCP-8, mMCP-11, the human tryptase and possibly also the mast cell and basophil specific carboxypeptidase A3 and the role of their targets in the recruitment of inflammatory cells and other immune-related functions at the area of parasite feeding. CPA3 is different from the other enzymes listed here above due to that it is an exopeptidase that removes one or a few amino acids from the C-terminal of a protein [53,54].

This is compared to the other enzymes that are endopeptidases that cleave inside proteins and thereby have a higher inactivating activity on larger proteins. CPA3 does thereby most likely have a minor role in the inactivation of the anticoagulants and other larger proteins, but may instead be involved in the regulation of peptide hormones and possibly also chemokines at the site of parasite infestation.

## 4. Materials and Methods

### 4.1. Recombinant Proteins

The coding regions for the different anticoagulants were extracted from the NCBI and Uniprot databases or from publications. The tick anticoagulant (TAP), the hirudin and the anophelin were all designed for the expression in bacterial cells. Accession numbers for TAP (P17726), hirudin (P09944) and anophelin [8]. The coding regions were after design ordered from Genscript (Piscataway, NJ, USA) as designer genes. After synthesis and sequence confirmation they were delivered in the pUC57 vector. The inserts were transferred to the bacterial expression vector pET21a by cleavage with the restriction enzymes BamHI and SalI and ligated into the expression vector. The vector for TAP also contains the coding region for the *E. coli* thioredoxin gene (Trx). Using Trx as a fusion partner is aimed to increase the solubility of the recombinant protein. The clones for hirudin and anphelin only contained an His_6_ tag and an enterokinase position N-terminally of the coding region for the anticoagulants. The clones were sequence-verified and transformed into *E. coli Rosetta gami* for protein expression. The tick serpin IRS-2 [18] (accession numberABI94056.2) was ordered from Genscript (Piscataway, NJ, USA) as a designer gene for expression in mammalian cells. Genscript also ligated the insert into the mammalian expression vector pCEP-Pu2 [55]. The IRS-2 also contains an N-terminal His_6_ tag but no enterokinase site. Following the transfer of the vector to *E. coli* by the transformation the vector was plasmid purified and sterilized by ethanol precipitation for transfection into HEK-293-EBNA cells. After transfection, the cells were selected for puromycin resistance and after a few weeks of culture, the conditioned cell medium was used to purify the recombinant protein on Ni^2+^ chelating IMAC column (Qiagen, Hilden, Germany).

### 4.2. Cleavage Reactions

Approximately 1 µg of recombinant TAP, hirudin, or anopheline was added to each cleavage reaction. The total volume was set to 13 µL in order to accommodate the final volume after the addition of sample buffer to one well on the gel. The reactions were run in 1× Phosphate buffered saline (PBS) pH 7.3. The cleavage was performed for 2.5 h at 37 °C. The reaction was terminated by the addition of 3 µL of 4× sample buffer. 0.5 µL β-mercaptoethanol was then added to each sample followed by heating for 5 min at 85 °C and then analyzed on 4–12% pre-cast gradient SDS-PAGE gels (Novex, Invitrogen, Camarillo, CA 93012, USA). The gels were stained overnight in colloidal Coomassie staining solution and de-stained with 25% of methanol for at least 3 h and then with H_2_O until the background is clear [56].

### 4.3. Chromogenic Substrate Assay

Chromogenic substrate assays were performed to determine the inhibitory activity of hirudin and anopheline on Thrombin to see that the recombinant anticoagulant proteins were biologically active. We used the chromogenic substrate Suc-GPR-pNA from Bachem (Bubendorf, Switzerland) to study the activity of thrombin, which displays tryptic activity with a preference for Arg in the P1 position. To study the activity of the human mast cell chymase, hamster chymase and opossum chymase we used a chromogenic substrate design for chymotrypsin like enzymes the substrate Suc-AAPF-pNA from Bachem (Bubendorf, Switzerland). Reactions were prepared in 96-well microtiter plates, to which 5 µL of substrate (0.2 mM final concentration), a few µL activated enzyme depending on activity, and PBS was added to a final volume of 200 µL. The reaction was done at 20 °C with measurements taken spectrophotometrically with a Versa-max microplate reader (Molecular Devices, Sunnyvale, CA, USA) at 405 nm at 0, 20, 40, 60, 120, 180, 240, 300 and 360 min. Reactions were run in triplets together with a blank to which no enzyme was added. Results were then graphed by subtracting the blank measurement at each time point and using the mean of the three reactions.

### 4.4. Cleavage Reactions of Cytokines and Chemokines with mMCP-8

Fifty-two recombinant mouse cytokines and chemokines were purchased from Immuno Tools (Friesoythe, Germany). The cytokines and chemokines were dissolved in PBS or sterile water, according to the recommendations of the supplier, to get an approximate concentration of 0.13 µg/µL. Subsequently, 13 µL (~1.7 µg) of the cytokine was mixed with 2 µL of the recombinant mMCP-8 (~400 ng) and incubated for 2.5 h at 37 °C. Two µL of PBS was used as control. The cleavage was performed in 1× PBS at pH 7.3. After incubation, the reactions were stopped with the addition of 3 µL of 4× sample buffer. Beta-mercaptoethanol (0.5 µL) was then added to each sample followed by heating for 7 min at 85 °C. The reaction mixtures were then analyzed on 4–12% pre-cast SDS-PAGE gels (Novex, Invitrogen, Camarillo, CA 93012, USA). To visualize the proteins, the gels were stained overnight in colloidal Coomassie staining solution and de-stained with 25% (*v*/*v*) methanol in ddH_2_O for 4 h [56]. The analysis was completely repeated and the result was essentially identical between the two independent experiments.

## Figures and Tables

**Figure 1 ijms-22-12627-f001:**
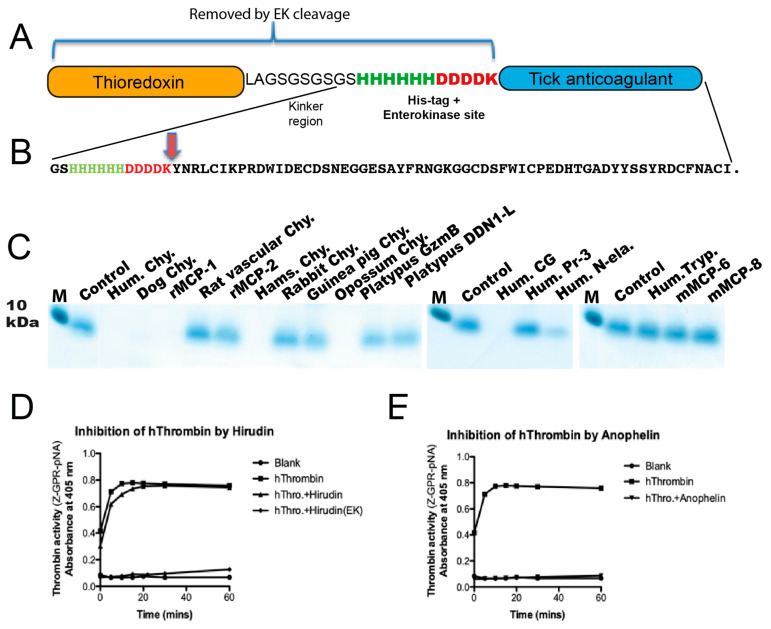
**Cleavage of the tick anticoagulant TAP protein by a panel of mammalian hematopoietic serine proteases.** Panel (**A**) shows the schematic structure of the recombinant Trx-6His-EK-TAP fusion protein. Panel (**B**) shows the sequence of the linker region between the Trx and the TAP coding regions containing the six-histidine affinity tag and the enterokinase site (EK), together with the entire sequence of the tick anticoagulant protein (TAP). The efficiency in cleavage of TAP was assayed in 15 µL samples with approximately 1 µg of TAP and varying amounts of protease depending on activity as determined previously. All proteases except mMCP-8 have been described in previous publications (Human chymase [30], Dog chymase [31], rMCP-1 [32], Rat vascular chymase (RVC) [33], rMCP-2 [34], Hamster chymase [35], Rabbit and Guinea pig Leu-ases [29], Opossum chymase [36], Platypus chymases [37], Human cathepsin G [38], Human proteinase 3 and N-elastase [39], Human mast cell tryptase and mMCP-6 [40]. Panel (**C**) shows the result of the cleavage reactions with the recombinant TAP and the various proteases. M marks the size marker for 10 kDa protein. Panel (**D**) shows a chromogenic substrate assay analyzing the inhibitory activity of hirudin on human thrombin with its chromogenic substrate Z-GPR-pNA. Panel (**E**) shows a chromogenic substrate assay analyzing the inhibitory activity of anophelin on human thrombin using the same chromogenic substrate as for hirudin.

**Figure 2 ijms-22-12627-f002:**
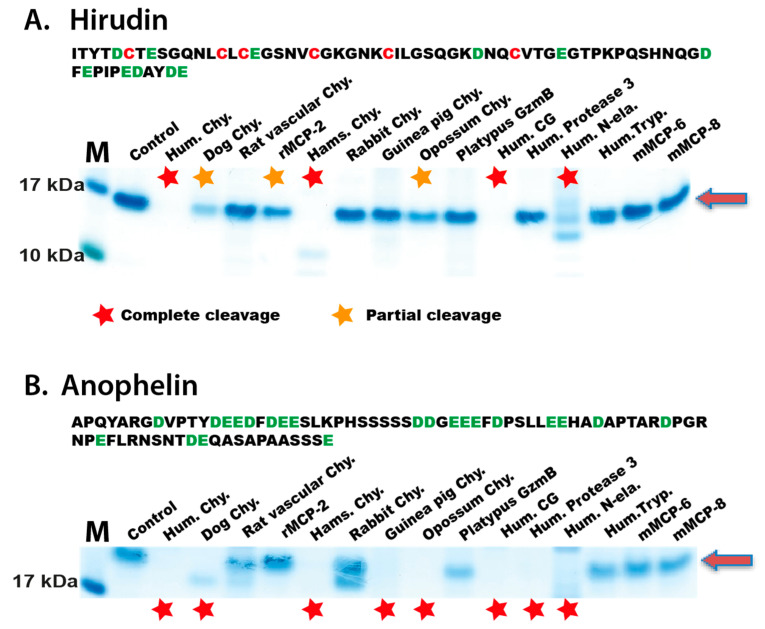
**Cleavage of the anticoagulant proteins, hirudin and anophelin, by a panel of mammalian hematopoietic serine proteases**. The efficiency in cleavage of three anticoagulant proteins were assayed in a 15 µL sample volume with approximately 1 µg of the anticoagulant protein and varying amount of protease depending on the activity as determined previously. All proteases except mMCP-8 have been described in previous publications (Human chymase [30], Dog chymase [31], rMCP-1 [32], Rat vascular chymase (RVC) [33], rMCP-2 [34], Hamster chymase [35], Rabbit and Guinea pig Leu-ases [29], Opossum chymase [36], Platypus chymases [37], Human cathepsin G [38], Human proteinase 3 and N-elastase [39], Human mast cell tryptase and mMCP-6 [40]. Panel (**A**) shows the amino acid sequence of hirudin where the cysteines have been marked in red and all negatively charged residues in green. Below the amino acid sequence, the cleavage reactions are presented for the panel of hematopoietic serine proteases. A red arrow marks the position of the hirudin protein on the gel. Panel (**B**) shows the amino acid sequence of anophelin where all negatively charged residues are in green. Anophelin contains no cysteines. Below the amino acid sequence, the cleavage reactions are presented for the panel of hematopoietic serine proteases. A red arrow marks the position of the anophelin protein. Red stars mark complete cleavage and orange stars partial cleavage.

**Figure 3 ijms-22-12627-f003:**
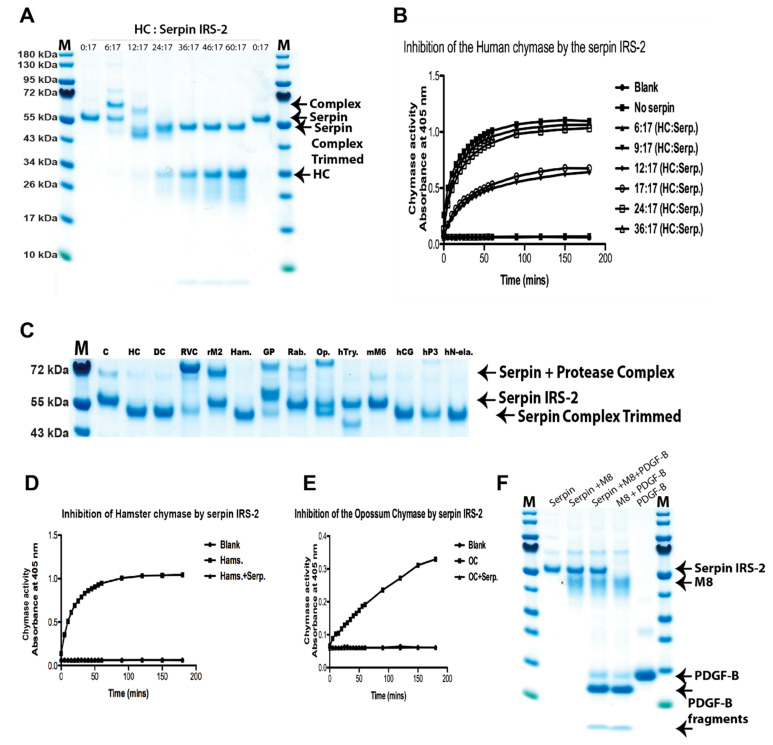
**Inhibiting activity of the tick serpin IRS-2 on a panel of mammalian hematopoietic serine proteases.** Panel (**A**) shows a titration of the activity of tick IRS-2 on human mast cell chymase. The amount of the recombinant tick IRS-2 was constant in all lanes. The amount of human chymase increased from left to right, starting and ending with two lanes with no chymase to show the size and purity of the recombinant IRS-2 and a reference point to evaluate the different complexes and cleavage products that appear following the interaction between the serpin and the enzyme. The ratio is then presented as follows. No chymase and 17 molar equivalents IRS-2 (0:17), six molar equivalents chymase and 17 equivalents IRS-2 (6:17) and until (60:17). The covalent serpin-chymase complex, the serpin, the trimmed serpin chymase complex and the unbound chymase are all marked by arrows to the right side of the panel (**A**). Panel (**B**) shows a chromogenic substrate assay of the different samples shown in Figure 3A using a chromogenic substrate (Suc-AAPF-pNA) for chymotrypsin-like proteases. In a ratio where the remaining free chymase starts to appear in panel (**A**) (12:17), cleavage of the substrate is also starting. Panel (**C**) shows an analysis of the inhibiting activity of a panel of hematopoietic serine proteases ranging from human to platypus enzymes, the same as in Figure 2, except mMCP-8, which is analyzed in panel (**F**). All proteases in this panel are inhibited by the serpin except for the rabbit Leu-ase and the tryptases, human tryptase and mMCP-6. The covalent serpin-enzyme complex, the serpin and the trimmed serpin protease complex are marked by arrows to the right side of the panel (**C**). Panels (**D**,**E**) show the activity of the serpin on hamster and opossum chymases using a chromogenic substrate. The panels also indicate that the trimmed complexes, seen in panel (**C**), are enzymatically inactive due to the serpin that is covalently bound to the active site of the enzymes. Panel F shows the potential interaction between the serpin and mMCP-8. mMCP-8 is not inhibited by IRS-2 and can cleave the substrate mouse PDGF-B even in the presence of the serpin. The serpin, mMCP-8, uncleaved mouse PDGF-B, and the two fragments of mouse PDGF-B generated after cleavage by mMCP-8 are all marked by arrows to the right side of the panel (**F**).

**Figure 4 ijms-22-12627-f004:**
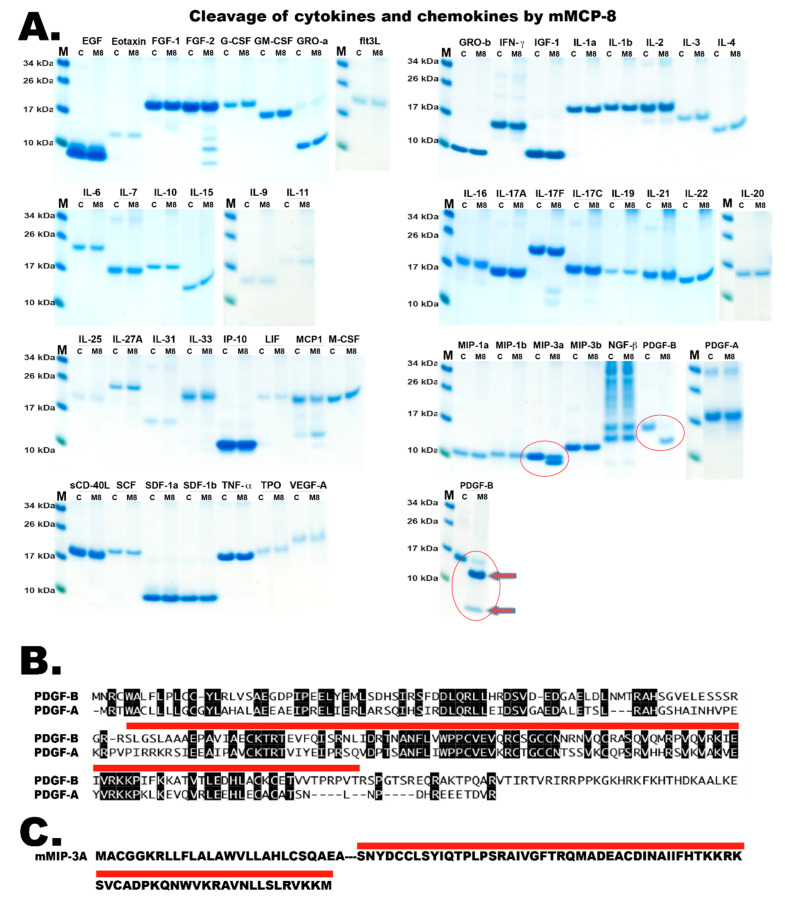
**Cleavage of cytokines and chemokines by mMCP-8.** Panel (**A**) shows the activity of recombinant mMCP-8 on a panel of 52 different recombinant mouse cytokines and chemokines. The name of the different cytokines and chemokines are listed above each lane, with one control sample without enzyme and one with mMCP-8. The cleavage was performed at 37 °C for two and a half hours after which sample buffer was added and the sample was separated on 4–12% gradient gels. Only two of these cytokines/chemokines (MIP-3a and PDGF-B) were cleaved by mMCP-8. These two panels are marked with red circles. One additional panel with the cleavage of PDGF-B is presented where the gel was run a shorter time to also visualize the second shorter band generated after cleavage. This panel is also marked with a red circle and two arrows pointing at the cleavage products generated by mMCP-8 cleavage. MIP-3a is cleaved close to its N or C-terminal ends whereas PDGF-B is cleaved further into the protein. Panel (**B**) shows a sequence comparison between mouse PDGF-A and PDGF-B using the DNASTAR program and Megaline. All amino acid residues identical between the two proteins are marked with a black background. The red line marks the sequence of the active fully processed mouse PDGF-B. Panel (**C**) shows the amino acid sequence of mouse MIP-3a. The red line marks the sequence of the active fully processed mouse MIP-3a.

## Data Availability

All data has included in the manuscript.

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
