# Peer review of "Mast Cells and Basophils in the Defense against Ectoparasites: Efficient Degradation of Parasite Anticoagulants by the Connective Tissue Mast Cell Chymases"

_ijms, 2021, doi:10.3390/ijms222312627_

Round 1
Reviewer 1 Report
Host protection against ectoparasites is important and the contribution of the immune cells still leave much to be explored. One potential way for the host to protect itself from these parasites is inactibvation their anticoagulants.Presented work by Hellman and colleagues contribute relevant information on this topic. However, there are some minor considerations:
- The Introduction is too extensive. There are some unnecessary repetition in the Introduction and Disscusion.
- I regret that charts with these legend symbols are unreadable.
- Abbreviations should be used consistently.
Author Response
Reviewer 1.
The Introduction is to extensive and there are repetitions.
- The only repetition we can find is that we mention ticks, mosquitos and leeches twice. We have now removed the second time. We are little trapped in between the two reviewers concerning the length of the introduction reviewer 1 want it shorter and reviewer two want additions to the introduction. We feel that we have kept it to a minimum not to make it too long but still giving background to all the individual parts that are included in the article from the anticoagulants, over the inhibitors and then giving a short description of the hematopoietic serine proteases that the entire article is centered around. So we have difficult in seeing how to make it shorter without affecting the understanding of the subject. Sorry. We have also looked at the discussion and what may seem as repetitions is just to refresh the reader to the subject of what´s being discussed. The second reviewer did also not seem to think the discussion was to long but rather lacked discussion of certain aspects of other mast cell proteases. We have now added such text actually making the discussion slightly longer but also removed one misplaced sentence.
- I looked through the figures and all symbols look fine in my version of the manuscript.
- I have also checked the abbreviations and I cannot find where we are inconsistent. Very sorry. TAP and IRS-2 are used consistently and mMCPs are used consistently.
Reviewer 2 Report
The authors carried out an interesting exploratory work on the defense mechanisms of mast cells and basophils against the anticoagulants of ectoparasites, with particular emphasis on mast cell chymase. The results obtained expand the existing understanding of the mechanisms of participation of mast cells in innate immunity, and provide a holistic picture in phylogenetic terms. However, despite the data obtained, the reviewer had a number of remarks that must be taken into account in the further preparation of the article for publication.
Remarks:
Line 119: "We show that the connective tissue mast cell chymases, but not the closely related mucosal mast cell chymases, efficiently cleaved all three tested anticoagulants."
1.1. How can the authors explain the difference in the functional activity of chymase in the mast cells of the connective tissue and in the mast cells of the mucous membranes? As is known, humans have one gene for chymase CMA1 (Pejler G, et al., 2007, Pejler G., et al., 2010), and its presence in mast cells characterizes the characteristics of the protease phenotype of mast cells depending on the content of specific proteases. In addition, on line 121, the authors write: "These results suggest an important role for the mast cell-specific chymases in our defense against ectoparasite infections."
1.2. It is known that mast cells have different levels of A3 carboxypeptidase according to histotopographic characteristics (Pejler G, et al., 2009, Siddhuraj P et al., 2021). Including known different content of CPA3 in mucous and connective tissue mast cells. Why do the authors not take this into account when carrying out their work, since the degrading effects of chymase in relation to toxins may be due to the actions of exopeptidases, including CPA3 (Pejler G, et al., 2009)? This issue should be taken into account in the "Discussion" section.
2.
Line 153: “The only connective tissue proteases that did not cleave the tick anticoagulant protein were the two platypus enzymes (Platypus GzmB and Platypus DDN1) and the mast cell enzymes from rabbit and guinea pig and (Rabbit Chy. And Guinea pig Chy.) ( Figure 1C). "
- What features of enzymes do the authors associate the discovered biological effects with?
3.
The authors investigated the direct biological effect of mast cell proteases in relation to a number of substances that prevent blood clotting. How objective is it possible to transfer the obtained data to analogous reactions in vivo? How does the concentration of the enzymes used in the experiment compare with the actual level of chymase and other proteases in the specific tissue microenvironment of various organs? In addition, secreted specific mast cell proteases can complement each other's action on the degradation of certain proteins, in particular, chymase and carboxypeptidase A3 (Pejler G, et al., 2009).
Recommendation:
In the discussion, it is desirable to present the mechanism of the development of the effect of specific mast cell proteases, considering the aspects of the reception of ectoparasite toxins entering the connective tissue, the mechanisms of secretory pathways of mast cell proteases, the mechanisms of maintaining certain levels of enzymes in the tissue microenvironment, as well as enzymatic mechanisms that ensure the neutralization of anticoagulants in the extracellular matrix. In particular, a number of review articles are known, on the basis of which it is possible to suggest variants of chymase secretion with maintenance of a certain level in the extracellular matrix (Blank U et al., 2014, Vukman KV et al., 2017, Atiakshin et al., 2019, 2021).
Useful literature:
Pejler G, Abrink M, Ringvall M, Wernersson S. Mast cell proteases. Adv Immunol. 2007;95:167-255. doi: 10.1016/S0065-2776(07)95006-3. PMID: 17869614.
Pejler G, Knight SD, Henningsson F, Wernersson S. Novel insights into the biological function of mast cell carboxypeptidase A. Trends Immunol. 2009 Aug;30(8):401-8. doi: 10.1016/j.it.2009.04.008. Epub 2009 Jul 28. PMID: 19643669.
Pejler G., Rönnberg E., Waern I., Wernersson S. Mast cell proteases: multifaceted regulators of inflammatory disease. Blood. 2010;115(24):4981–4990.
Blank U, Madera-Salcedo IK, Danelli L, Claver J, Tiwari N, Sánchez-Miranda E, Vázquez-Victorio G, Ramírez-Valadez KA, Macias-Silva M, González-Espinosa C. Vesicular trafficking and signaling for cytokine and chemokine secretion in mast cells. Front Immunol. 2014 Sep 22;5:453. doi: 10.3389/fimmu.2014.00453. PMID: 25295038; PMCID: PMC4170139.
Vukman KV, Försönits A, Oszvald Á, Tóth EÁ, Buzás EI. Mast cell secretome: Soluble and vesicular components. Semin Cell Dev Biol. 2017 Jul;67:65-73. doi: 10.1016/j.semcdb.2017.02.002. Epub 2017 Feb 9. PMID: 28189858.
Atiakshin D, Buchwalow I, Tiemann M. Mast cell chymase: morphofunctional characteristics. Histochem Cell Biol. 2019 Oct;152(4):253-269. doi: 10.1007/s00418-019-01803-6. Epub 2019 Aug 8. PMID: 31392409.
Atiakshin D, Buchwalow I, Horny P, Tiemann M. Protease profile of normal and neoplastic mast cells in the human bone marrow with special emphasis on systemic mastocytosis. Histochem Cell Biol. 2021 May;155(5):561-580. doi: 10.1007/s00418-021-01964-3. Epub 2021 Jan 25. PMID: 33492488; PMCID: PMC8134284.
Siddhuraj P, Clausson CM, Sanden C, Alyamani M, Kadivar M, Marsal J, Wallengren J, Bjermer L, Erjefält JS. Lung Mast Cells Have a High Constitutive Expression of Carboxypeptidase A3 mRNA That Is Independent from Granule-Stored CPA3. Cells. 2021 Feb 3;10(2):309. doi: 10.3390/cells10020309. PMID: 33546258; PMCID: PMC7913381.
Author Response
Reviewer 2.
- Human connective tissue mast cells are the only mast cell in human that express a chymase and they have both the chymase and cathepsin G. In contrast in rodents both types of mast cells express chymases. In the mouse CTMC express mMCP-4 and MMC express mMCP-1 and mMCP-2. mMCP-2 is catalytically inactive so the only chymasse of importance for MMCs is mMCP-1 which is highly active. We show the activity of the rat counterpart of mMCP-1 in this article namely rMCP-2 which does not seem to have any activity on these anticoagulants. The rat vascular chymase is also inactive against the anticoagulants and they show slightly different cleavage specificity compared the connective tissue proteases. The connective tissue proteases show a preference for negatively charged residues in the P2’ position which the mucosal mast proteases lack the mucosal mast cell proteases have a stronger preference for Serine in the P1´position compared to the connective tissue proteases. These two differences are probably strongly connected to the difference in cleavage activity we see against these anticoagulants.
- CPA3 is a carboxypeptidase and removes only a few amino acids from the C-terminal of a protein and is therefore most likely not involved in inactivating larger proteins like these anticoagulants. The primary effect of CPA3 has been on peptides like endothelin, where CPA3 can have a major effect. We therefor think CPA3 most likely have a minor role on the proteins we look at here but may have a major impact on peptide hormones possibly on some chemokines in the area of infection. Concerning the role of the rabbit and guinea pig Leu-ases. We still have no good explanation and clue to their potential targets and why these two species lack a classical mast cell chymase. We try to give a short explanation to the lack of cleavage by the platypus enzymes, by saying that they are water dwelling and that they live in a part of the globe that has a very different fauna that may be the reason for the lack of cleavage activity.
- The concentration of these enzymes in vivo are very relevant but also very difficult to determine. They also change a lot with the increasing distance from the cell. Close to the mast cell the concentration is very high and the further we go from the cell the concentration drops. The activity also depends on the presence of protease inhibitors and on the heparin concentration. If blood vessels open then the concentration of inhibitors quickly increase resulting a rapid drop in activity of the enzymes. So we think that close to the cell the activity is high but then drop quickly by distance why we in the same tissue will have a number of different conditions. However, what we can conclude is that when the enzyme is there and active it can efficiently cleave these three anticoagulants.It is true that the enzymes can complement each other as seen in the case of endothelin where the chymase cleaves in the middle but does not inactivate the peptide. However, when CPA3 removes a few amino acids from the C-terminal it kills the activity of this vasoactive peptide. However, as mentioned above we think CPA3 has little effect on these anticoagulants as they most likely still are active after removal of a few amino acids from the C-terminal why endopeptidases most likely are the most important of the mast cell enzymes in this respect. We have added a short section in the end of the discussion (marked in red) concerning this issue. We have also added a short section on other triggers of mast cell activation such as substance P activating MRGPRX2 and the anaphylotoxins that can become activated in the area of ectoparasite attack. We have now added two references on CPA3 in the end of the manuscript.